# Origins and Bioactivities of Natural Compounds Derived from Marine Ascidians and Their Symbionts

**DOI:** 10.3390/md17120670

**Published:** 2019-11-28

**Authors:** Xiaoju Dou, Bo Dong

**Affiliations:** 1Laboratory of Morphogenesis & Evolution, College of Marine Life Sciences, Ocean University of China, Qingdao 266003, China; douxiaoju@stu.ouc.edu.cn; 2Laboratory for Marine Biology and Biotechnology, Qingdao National Laboratory for Marine Science and Technology, Qingdao 266237, China; 3Institute of Evolution & Marine Biodiversity, Ocean University of China, Qingdao 266003, China; 4College of Agricultural Science and Technology, Tibet Vocational Technical College, Lhasa 850030, China

**Keywords:** ascidian, symbiotic microbes, natural compounds

## Abstract

Marine ascidians are becoming important drug sources that provide abundant secondary metabolites with novel structures and high bioactivities. As one of the most chemically prolific marine animals, more than 1200 inspirational natural products, such as alkaloids, peptides, and polyketides, with intricate and novel chemical structures have been identified from ascidians. Some of them have been successfully developed as lead compounds or highly efficient drugs. Although numerous compounds that exist in ascidians have been structurally and functionally identified, their origins are not clear. Interestingly, growing evidence has shown that these natural products not only come from ascidians, but they also originate from symbiotic microbes. This review classifies the identified natural products from ascidians and the associated symbionts. Then, we discuss the diversity of ascidian symbiotic microbe communities, which synthesize diverse natural products that are beneficial for the hosts. Identification of the complex interactions between the symbiont and the host is a useful approach to discovering ways that direct the biosynthesis of novel bioactive compounds with pharmaceutical potentials.

## 1. Introduction

Marine ascidians are ancestral chordates belonging to the species of tunicate in the subphylum Urochordate [1]. Since they exhibit vertebrate-like tadpole characteristics in larval form but are invertebrate filter feeders in adult form, ascidians occupy a unique evolutionary position between vertebrate and invertebrate [2]. Such features are reflected in that they not only possess genes that do not exist in invertebrates, but they have retained functions that are not evolved in vertebrates. For instance, ascidians produce alternative oxidase and phytochelatins, which were thought to be protostome-specific [3]. They are also unique among the animal kingdom in their ability to biosynthesize cellulose [4,5].

There are around 3000 species of ascidians spread over the world with both asexual and sexual reproduction, forming diversified sizes and shapes [6]. Rapid evolution of tunicate genomes—with their short lifespan and the two modes of reproduction—is responsible for the diversity and the adaptability of ascidians, which reflect the structural novelty and the functional diversity of their secondary metabolites [7,8].

The first ascidian compound discovered was geranyl hydroquinone, which was isolated from *Aplidium* sp. in 1967 with the cytotoxicity against leukemia [9,10]. Afterward, discovery of ascidian- derived natural compounds started to grow considerably. There were only 230 secondary metabolites isolated from ascidians before 1992 [11], whereas, from 1994 to 2014, nearly 600 new metabolites were reported [12]. Since 2015, ascidians have been the source of more than 300 discovered metabolites [13,14]. To date, around 1200 structurally distinct, natural compounds have been identified from ascidians [14,15]. Ascidians provide rich sources of bioactive secondary metabolites and have yielded promising drugs or drug leads. According to the U.S. Food and Drug Administration (FDA) database, nine marine or marine-derived drugs have been approved for therapeutics proposes, among which ecteinascidin 743 (Yondelis^®^) from the ascidian symbiotic bacteria *Candidatus Endoecteinascidia* frumentensis, and dehydrodidemnin B (Aplidin^®^) from the ascidian *Aplidium albicans* are in usage for cancer treatments (Figure 1a,b) [16,17]. Three molecules out of 12 are in clinical trial evaluations (Figure 1c) [18]. Ascidian compounds range from derivatives of alkaloids to peptides, polyketides, quinones, and steroids, and they have diverse bioactivities, including antitumor, antiviral, and anti-inflammatory activities in vitro and in vivo (Figure 1d,e) [12,13,19,20].

Despite the various metabolites that are produced by ascidians, symbiotic microbes have also been found to be rich sources of bioactive and pharmacologically valuable compounds [21]. Some compounds first isolated from ascidians have been proven to be produced by symbiotic microbes. There is increasing evidence that around 100 (8%) of the known compounds isolated from ascidian symbionts offer a renewable supply, which is important for marine drug candidates (Figure 1f) [22,23,24]. 

In this review, we first document the novel compounds produced by the host animals, and then we focus on the existing metabolites that have been identified from ascidian symbionts. Subsequently, we discuss the roles of these symbionts in the biosynthesis of natural products. We also summarize the diversity of microbe communities, which not only benefit the hosts but are also involved in their regulatory roles in the biosynthesis of natural products. The interactions between ascidians and their symbionts indicate the existence of some small molecules that are directly recognized by symbiotic microbes to modulate and control the production of natural compounds, which, in turn, affect the host ascidians. 

## 2. Compounds from Ascidians and Their Symbionts

The evolutionary position of urochordate marine ascidians indicates their importance in drug discovery. Increasing evidence indicates that many bioactive compounds previously isolated from ascidians closely resemble microbe metabolites or are proposed to originate from microbes, since the single molecule governing their biosyntheses resides exclusively in microbes [21,25]. These findings suggest that ascidian-associated microbes are important sources of bioactive compounds. Therefore, symbiotic microbes offer a new perspective for marine drug developments [26]. The remarkably high chemical diversity of secondary metabolites from ascidians and symbionts, including alkaloids, polypeptides, polyketides, and other structural families, contributes to the development of new medicinal substances as promising drug candidates.

### 2.1. Alkaloids

Alkaloids have provided the majority of ascidian originating bioactive compounds. More than 70% of ascidian bioactive compounds are in the class of alkaloids (Figure 1d). Alkaloids generally represent a highly diverse group of compounds containing cyclic structures with at least one basic nitrogen atom being incorporated within. These compounds have been identified from diverse ascidians and display antimicrobial and anticancer activities via inhibiting the activities of kinases, including Protein Kinase B (PKB) and Cyclin-Dependent Kinases (CDKs) [27], interfering with topoisomerase (TOPO) I [28], altering the mitochondrial membrane potential [29,30], and binding to the DNA minor groove to inhibit transcriptional activation [31,32]. 

#### 2.1.1. Alkaloids from Ascidians

##### Didemnidines

Didemnidines A (**1**) and B (**2**) (Figure 2) are two indole spermidine alkaloids isolated from the New Zealand ascidian *Didemnum* sp. [33]. Both are active as inhibitors of phospholipase A2 and the farnesyltransferase enzyme without cytotoxicity. Didemnidine B also shows mild activity on the malaria parasite. Meanwhile, the two synthetic intermediates of didemnidines display moderate cytotoxicity toward L6 cells and inhibit the proliferation of parasites. The antiparasitic activity of didemnidine B provides the opportunity to explore the didemnidines family as antimalarial and antitrypanosomal agents [34].

##### Meridianins

Meridianins (Figure 3) are brominated 3-(2-aminopyrimidine)-indoles isolated from the ascidian *Aplidium meridianum* [35]. As they are structurally similar to variolins, meridianins are identified as a promising kinase-inhibitory scaffold, which inhibits various protein kinases, such as CDKs, glycogen synthase kinase-3, cyclic nucleotide-dependent kinases, and casein kinase. Meridianins also prevent cell proliferation and induce cell apoptosis, probably because of their interference with the activity of kinases, which are important for cell division [36]. Among the identified meridianins, meridianins B (**3**) and E (**4**) are thought to be the most potent in this class of compounds because of their considerable antitumor activities [37].

##### Herdmanines

Herdmanines are a series of nucleoside derivatives isolated from the ascidian *Herdmania momus*. Herdmanines A–D (5–8) (Figure 4) are found to inhibit the production and the expression of mRNA of pro-inflammatory cytokines, while herdmanines C (**7**) and D (**8**) have moderate suppressive effects on the production of Lipopolysaccharide (LPS)-induced nitricoxide [38,39].

As one of the major classes of secondary metabolites isolated from ascidians, alkaloids exhibit various pharmacological activities. The list of other alkaloids and their molecular targets are presented in Appendix A.

#### 2.1.2. Alkaloids from Ascidian-Associated Microbes

##### Ecteinascidins

Ecteinascidins belong to the tetrahydroisoquinoline alkaloid family, which was previously isolated from the ascidian *Ecteinascidia turbinata* [31]. Recently, it was isolated from the ascidian symbiotic bacteria *Ca.E.* frumentensis [40]. There are six known ecteinascidins (ecteinascidin 729, 743, 745, 759A, 759B, and 770) exhibiting potent antitumor activities [41]. Among them, ecteinascidin-743 (ET-743) (9) (Figure 5) shows striking activity against a variety of tumor cells. It binds with the minor groove of DNA and leads to the sequence-specific alterations in transcription. Its binding to DNA also triggers DNA cleavage, and then it causes double-stranded breaks as well as interruption of the cell cycle, apoptosis of cancer cells, and down-regulation of some transcription factors [42]. Beside this, ET-743 has modulatory effects on the tumor microenvironment. These effects are critically deemed to be important in cancer therapy because of the resultant inhibition of neoangiogenesis and the metastatic potential of cancer cells. With promising results in clinical trials of various chondrosarcoma, ET-743 is now in clinical use in more than 70 countries for treatment of cancer [41].

##### Eusynstyelamides

Eusynstyelamides A–C (10–12) (Figure 6) are alkaloids previously isolated from the ascidian *Eusynstye lalatericus*. However, the structurally similar eusynstyelamide B and ent-eusynstyelamide B, a secondary metabolite from a bryozoan species, suggests the compounds might be synthesized by symbiotic microbes. Eusynstyelamides A–C show specific cytotoxicity against neuronal nitric oxide synthase (nNOS) and show modest anticancer and antibacterial activities [43]. Eusynstyelamides A and B display inhibitory activities against *Staphylococcus aureus* and the plant regulatory enzyme pyruvate phosphate dikinase (PPDK) [43,44]. In addition, eusynstyelamide B exhibits anti-proliferation activity in MDA-MB-231 cells. It causes a strong cell cycle block at G2/M phases and induces cell apoptosis. Nitric oxide synthase family genes are regarded as its potential target [43,45]. The bioactivities of eusynstyelamides suggest their potential for further medical studies.

##### Sesbanimides

Sesbanimide A (13) was isolated from the bacteria *Agrobacterium,* which is associated with the ascidian *E. turbinata*, while sesbanimide C (14) was isolated from the bacteria *Agrobacterium*, which is associated with the ascidian *Polycitonidae* sp. (Figure 7) [46,47]. Sesbanimide A showed cytotoxicity against the growth of mouse leukemia cells and inhibited the proliferation of mouth epidermal carcinoma (KB) cell. However, the activity was approximately one-tenth that observed for sesbanimide C [47]. 

##### LL-14I352 α and β

LL-14I352 α (15) and β (16) (Figure 8) were isolated from LL-14I352, a bacterium associated with an unidentified ascidian from the Pacific Ocean. LL-14I352 α exhibits strong activity against gram-positive bacteria, but it shows weaker anti-gram-negative bacterial activity, while LL-14I352 β is less active on gram-positive bacteria than LL-14I352 α. Both compounds have been found to inhibit the growth of human ovarian and colon cancer cells. They also exhibit the activities of inhibiting DNA, RNA, and protein synthesis [48].

Except for the alkaloids discussed above, many other alkaloids have been isolated from diverse ascidian-associated microbes. Details on structures, host animals, origin microbes, and bioactivities of these compounds are summarized in Appendix A.

### 2.2. Polypeptides

Polypeptides have garnered increased interest because of their significant bioactivities. Around 5% of the identified ascidian originating compounds belong to peptides (Figure 1d). Peptides are one of the major structural classes isolated from ascidians, including linear peptides, depsipeptides, and cyclic peptides, with residue numbers spanning from two to forty eight [49]. Most of the active peptides from ascidians have complex cyclic or linear structures that are rarely found in terrestrial animals. Peptides isolated from ascidians and their associated microbes have opened a new perspective for pharmaceutical developments [50]. These peptides affect cell behavior with different mechanisms, including inducing apoptosis, affecting the tubulin–microtubule equilibrium [51], or inhibiting angiogenesis [52].

#### 2.2.1. Polypeptides from Ascidians

##### Vitilevuamide

Vitilevuamide (17) (Figure 9) is a bicyclic peptide isolated from the ascidians *Didemnum cuculiferum* and *Polysyncranton lithostrotum*. With a functionally correlated totaxol analog, vitilevuamide was shown to be active against P388 mouse lymphocytic leukemia in vivo [53]. The cytotoxic mechanism is due to its inhibition of tubulin polymerization without competitive inhibition of the vinblastine binding site, and it weakly affects GTP binding to tubulin as well as causes cell cycle arrest cells in the G2/M phases [53].

The cytotoxicity of vitilevuamide made it a viable candidate for in vivo experiments. With the potential ability to increase the life-span of leukemic mice, vitilevuamide has been approved by the FDA for preclinical tests [53,54].

##### Diazonamides

The diazonamides are a family of macrocyclic peptides isolated from the ascidian *Diazona angulata* [51]. Among the members of diazonamides, diazonamide A (18) (Figure 10) has been evaluated for its antitumor activities. The results indicate that it is a tubulin-binding agent and blocks the cell cycle in the G2/M period [55]. Notably, it does not compete with vinblastine or colchicine to bind with tubulin, while it interacts with ornithine-delta-amino transferase (OAT), a mitochondrial enzyme involved in spindle assembly and mitosis control. Diazonamide A is a potential chemotherapeutic agent without significant toxicities in animal models. It is, therefore, potentially regarded as an efficient agent in anticancer drug development [56]. 

##### Chondromodulin-1 (ChM-1)

ChM-1, a 25 kDa glycoprotein originally isolated from fetal bovine cartilage, could be converted to a 12 kDa mature peptide at the C-terminal RERR amino acid cleavage site and subsequently secreted to the extracellular matrix [57]. Recently, ChM-1 was identified from the invertebrate urochordate ascidian *Ciona savignyi* [58]. As an invertebrate animal without cartilage and vascellum, the role of the highly expressed ChM-1 indicated its novel function. In vitro cultured cell experiments show that the ascidian-originated ChM-1 mature peptide (Cs-mChM-1) presents dual roles in different cell types—it promotes the proliferation of mouse osteoblastic cells (MC3T3-E1) and protects H_2_O_2_ oxidative injury. In human neuroblastoma (SHSY5Y) cells, human cervical cancer (HeLa) cells, and human umbilical vein endothelial cells (HUVECs), Cs-mChM-1 suppresses cell proliferation and inhibits angiogenesis of HUVECs. Further experiments revealed that Cs-mChM-1 modifies cell behavior through regulating the cell cycle and cell adhesion [58]. These results suggest that Cs-mChM-1 is a potential antioxidant and antitumor agent.

##### CS5931

CS5931 is a polypeptide from *C. savignyi* with a molecular weight of 5931 Da. Sequence analysis reveals a high structural homology between CS5931 and human granulin A [59]. CS5931 exhibits significant cytotoxicity on several cancer cell types and induces apoptosis via a mitochondrial-mediated pathway. Cell cycle analysis demonstrates that CS5931 causes cell cycle arrest at the G2/M phases. Further studies found out that it inhibits the proliferation, the migration, and the formation of capillary-like structures of HUVECs, represses spontaneous angiogenesis of the zebrafish vessels, and blocks the production of vascular endothelial growth factor (VEGF). Moreover, CS5931 also reduces the expression of matrix metalloproteinases (MMP-2 and MMP-9) at both protein and mRNA levels in HUVECs. Enolase 1 was then identified as its molecular target [60,61]. These observations clearly demonstrate that CS5931 achieves antitumor activities both in vitro and in vivo, and it possesses the potential for therapeutic application.

The details of other polypeptides extracted from ascidians are listed in Appendix A.

#### 2.2.2. Polypeptides from Acidian-Associated Microbes

##### Didemnins

The didemnins are cyclic depsipeptides containing a macrocyclic core and consist of an isostatine and a -(-hydroxyisovaleryl) propionyl (Hip) group. Didemnins were first obtained from the ascidian *Trididemnum solidum* in 1981 [62], and they subsequently were proven to be produced by the ascidian-associated microbes α-proteobacteria *Tistrella mobilis* and *Tistrella bauzanensis* [63].

Didemnins A–C (19–21) (Figure 11) were initially identified from ascidian symbiotic microbes. Among them, didemnin B (20), with a high cytotoxicity activity, was the first marine natural product to be utilized in clinical trials [64].

Several studies have shown that didemnin B suppresses the proliferation of cancer cells by acting as a cell-cycle arrest agent. G1/S phase cells are more sensitive to didemnin B. It is known that the antiproliferative potency of didemnin B is due to interference with the mitogenic signal transmission, such as the inhibitors of kinases, phosphatases, and elongation factors [65,66].

The high efficiency of the antitumor activity of didemnin B provided the ground for clinical trials. Unfortunately, the trials were stopped because of cardiac and neuromuscular toxicities [67]. However, the structurally similar compound dehydrodidemnin B (aplidine) isolated from the Mediterranean tunicate *Aplidium albicans* was shown to be more potent and less toxic. It is currently being evaluated in phase II and III trials for the treatment of medullary thyroid carcinoma, renal-cell carcinoma, melanoma, and tumors of neuroendocrine origin [68,69]. The subtle structural difference of didemnin B and dehydrodidemnin B causes critically functional changes, which lead to different clinical application profiles. 

##### Patellamides

The patellamides are cyclic peptides isolated from the cyanobacterium *Lissoclinum patella*, with host animals belonging to the ascidian Didemnidae family [70]. Patellamides A, C, and D (22–24) (Figure 12) exhibit cytotoxic effects. Patellamides A (22) and C (23) inhibit the growth of L1210 murine leukemia cells, while patellamide D (24) acts as a resistance-modifying agent in the multidrug-resistant human leukemic cell lines, indicating its potential as a drug-resistance modulator [71].

Most of the peptides discovered from ascidian-associated microbes contain circular sequences and exhibit antibiotic or antitumor activities. Appendix A lists a series of peptides isolated from ascidian symbiotic microbes.

### 2.3. Polyketides

#### 2.3.1. Polyketides from Ascidians

Polyketides are the other important compounds in the screening of secondary metabolites from ascidians [around 80 of 1200 (7%)] (Figure 1d). Polyketides are complex molecules built from simple carboxylic acids and synthesized by polyketide synthetases [72]. Polyketide synthetases are large multienzyme machineries. Natural products having polyketide and non-ribosomal peptide (NRP) structures are generally found to be of microbial origin [13]. Polyketides and the synthetic analogs have been discovered as important lead compounds with various activities, such as blocking protein tyrosine phosphatase and inhibiting the ATP synthase complex [73].

##### Palmerolide A

Palmerolide A (25) (Figure 13) is a macrocyclic polyketide isolated from the ascidian *Synoicum adareanum*. Palmerolide A displays selective cytotoxicity toward melanoma by inhibiting V-ATPase [74]. As a promising compound, palmerolide A was chemically synthesized by enantio-selective methods [75], by which a large amount of palmerolide A and its analogs could be made for the investigation of biological function and application.

##### Mandelalides

Mandelalides are variously glycosylated polyketides isolated from the ascidian *Lissoclinum* sp. Mandelalides A and B show potent cytotoxicity to NCI-H460 cells and mouse Neuro-2a neuroblastoma cells [76]. Mandelalide B also displays a potent antifungicidal activity against *Candida albicans* [77]. The inaccessible supply of the source of promising biomedical actives indicates the importance of total chemical syntheses of the valuable compounds.

##### Phosphoeleganin

Phosphoeleganin is a novel phosphorylated polyketide from the ascidian *Sidnyum elegans.* Phosphoeleganin has no significant cytotoxicity against human prostate cancer (DU145) cells and human breast cancer (MCF-7) cells, but it shows inhibitory activity against the protein tyrosine phosphatase 1B [73]. This compound is expected to be a new hit for the treatment of diabetes and obesity.

Polyketides from marine ascidians have become important sources in drug discovery. Appendix A provides several polyketides based on their bioactivities.

#### 2.3.2. Polyketides from Ascidian-Associated Microbes

##### Patellazoles

Patellazoles A–C (26–28) (Figure 14) are a family of compounds produced by the α-proteobacterium *Candidatus Endolissoclinum faulkneri*, a microbe associated with the ascidian *L. patella*. These compounds display cytotoxicity towards HCT 116 cells by inhibiting protein synthesis, arresting cell cycle at G1/S phases, and inducing cell apoptosis [78]. It is known that their inhibition effects on protein synthesis are achieved by interfering with the mTOR/p70 pathway [78]. 

##### Arenimycin

Arenimycin (29) (Figure 15) was isolated from the ascidian *E. turbinate*-derived bacteria *Salinispora arenicola*. Arenimycin inhibits the division of HCT cells as well as exhibits potent antimicrobial activities against the drug-resistant strains *Staphylococci* and *Mycobacterium* [79]. The antiproliferation and antibacterial activities make arenimycin a potential candidate for clinical medicine [80].

Ascidian symbionts are known as the true sources of bioactive polyketides. Polyketides originating from symbionts and their biological properties are provided in Appendix A.

### 2.4. Other Types of Compounds from Ascidians and the Host-Associated Microbes

Appendix A list the other classes of compounds from the host ascidians and their associated microbes with the biological activities and molecular targets, respectively.

## 3. The Effects of the Interaction between Ascidian-Associated Microbes and Hosts on the Production of Natural Compounds 

As filter feeders, ascidians harbor diverse microbes and are regarded as the ideal model for the study of marine eukaryote–microbe associations [81]. Various proteobacteria, cyanobacteria, and fungi reside in the extra- and the intracellular space of ascidians, many of which may function as beneficial symbionts or persist through host developmental events [82]. The potential role of microbes in ascidian biology varies from mutualistic symbiosis to nutritional sources. Microbes not only serve as food or enrich the diet of their hosts by fixing carbon and nitrogen, but they also are involved in the synthesis of natural products [83].

### 3.1. Proteobacteria

The phylum Proteobacteria is one of the main ascidian-associated microbes that produce bioactive metabolites. α-Proteobacteria and γ-Proteobacteria are the two dominant groups. The majority of these produced bioactive compounds exhibit antimicrobial, antitumor, and anticancer properties. *Ca. E.* frumentensis is one of the proteobacteria residing in the host ascidian *E. turbinate*, which produces a large number of tetrahydroisoquinolines, including ET-743 [40].

The similarity of ET-743 to the bacterial-derived natural products, such as saframycin A and safracin B, suggests that it is of prokaryotic origin [84,85]. Analysis of the ~631 kb *Ca. E.* frumentensis genome further demonstrates the existence of biosynthetic genes as well as the biosynthetic enzyme of ET-743 [86]. Metaproteomic analysis reveals that the genome of *Ca. E.* frumentensis related to production of primary metabolism is reduced, as it is missing the genes involved in early glucose catabolism but encodes several sugar phosphate transporters, and it lacks a number of key amino acids and cofactors, including coenzyme A (CoA), but it encodes genes linked to transporter functions [86,87]. These features suggest that proteobacteria require some essential metabolite or their precursors from the host [88]. The genome of the microbe is also missing a number of genes involved in peptidoglycan and lipid A biosynthesis, which are incorporated in the vast majority of proteobacteria. The absence of these genes within the *Ca. E.* frumentensis highlights that the survival of the microbe strongly relies on the host animal [89,90].

Moreover, *Ca. E.* frumentensis is also missing a number of key biosynthetic genes of secondary metabolism involved in the production of ET-743 [86]. Gene candidates for enzymes that catalyze the formation of certain precursors to produce ET-743 remain to be identified, which suggests that genes involved in ET-743 biosynthesis might come from other microbes or that the production of these enzymes needs the interaction between the symbiotic microbes and the host ascidian [87,91].

### 3.2. Cyanobacteria

Cyanobacteria are the only prokaryotes thus far described that are obligate photosymbionts in chordates [92]. In tropical ascidians of the family Didemnidae, *Synechocystis trididemni* and *Prochloron* sp. are the most common cyanobacterial symbionts [93,94]. *S. trididemni* contains phycoerythrin pigments, which is typical for most of cyanobacteria, while *Prochloron* sp. is a unique photosymbiont possessing chlorophyll *a* and *b* [95]. The genera *Acaryochloris marina,* a remarkable photosymbiont with chlorophyll *d* (chl *d*) for photosynthesis, was isolated from the ascidian *L. patella* [96]. The presence of *A. marina* was also reported in the ascidians *L. patella*, *Diplosoma similis*, and *Diplosoma virens* [97]. A diversity assessment of the cyanobacterial community inhabiting didemnid ascidians from Bahamas Islands showed *Synechocystis* sp. in the tunic of ascidians *Trididemnum solidum* and *Trididemnum cyanophorum* as well as acaryochloris-like symbionts living with ascidians *Lissoclinum fragile* and *L.* aff. *fragile*. The host identity strongly correlated with the identity of the photosymbionts found in the tunic [98,99].

Symbiont cyanobacteria have been found to locate both on the surface or in the cloacal cavities of the hosts, and they show high colony densities from larvae to adult [100]. Studies also reveal that the intact *Prochloron* cells containing vesicles or granules are interspersed in the tunic matrix of ascidian *Trididemnum miniatum* but without direct contact with the host cells [101]. However, the observation of a round cell mass derived from the remains of degenerative *Prochloron* in the ascidian *T. miniatum* tunic indicates that tunic phagocytes may dispose of defective photosymbionts and expel them from the colony [101].

As an oxyphototrophic prokaryote, *Prochloron* sp. is an extremely important nutrient source for its host ascidians through both photosynthesis and nitrogen fixation, which enable the host to occupy various environments [102].

Systematic screening of photosynthetic genes shows that *Prochloron didemni* possesses complete photosynthetic components and genes encoding the sole phycobiliprotein, which is regarded as a photoreceptor [103]. Another feature of the component is organization of *hli* genes encoding Scps proteins, hard light-induced proteins belonging to the plant Lhc protein family, which are involved in the absorption of excess excitation energy [104]. However, *P. didemni* contains a small number of genes involved in CO_2_ transport. The NADH dehydrogenase complex is encoded by homologues of *Synechocystis ndhD4* and *ndhF4* genes, which takes up CO_2_ and converts it to HCO_3_^-^ [105].

*Prochloron* sp. contains genes required to fix nitrogen. It also provides amino acids for the hosts by a nitrogenase complex, which is formed with two subunits: the Fe subunit coded by the *nifH* gene and the Mo-Fe subunit coded by the *nifD* and the *nifK* genes. The expression of *nif* genes and the process of nitrogen fixation are mainly affected by O_2_ and NH_3_ [106].

In addition, *Prochloron* sp. also protects the host against the toxicity from active forms of oxygen during photosensitizing processes [107]. The Cu-Zn metalloprotein in *Prochloron* sp. is a cyanide-sensitive superoxide dismutase, and it is not only involved in carbonic anhydrase reactivity and phosphoester hydrolysis for photosynthesis but is also required for oxygen activation [107,108,109]. Studies revealed the increasing activities of superoxide dismutase (SOD), ascorbate peroxidase, and catalase under high irradiance in ascidian *L. patella*-harbored *Prochloron* sp. [107]. It is still not clear whether UV radiation can penetrate ascidian host tissues, but the visible radiation reaching the symbionts in the tunic is decreased by 60 to 80% [102]. These findings provide good evidence that symbionts are important to prevent the host from the toxicity produced by active forms of oxygen by removing superoxide radicals (O_2_-), hydrogen peroxide (H_2_O_2_), and hydroxyl radicals (HO·) during photosensitizing processes [107]. The removal of these superoxide radicals is important in preventing the inactivation of ribulose-1,5-bisphosphate carboxylase/oxygenase, the primary CO_2_-fixing enzyme in *Prochloron* sp. [110], and in protecting the nitrogenase enzyme from inactivation by reactive oxygen species (ROS) [111].

Another protective effect of *Prochloron* sp. is to screen out UV radiation with mycosporine-like amino acids (MAAs), substances with absorption maxima ranging from 310 to 360 nm [112]. In high-ultraviolet (UV) tropical environments, *Prochloron* sp. specifically localizes in tunic bladder cells of the ascidians and employs MAAs to prevent the host from excess UV irradiation [113]. Genome data indicate that MAAs are produced by the symbiont and are transported to the animal [114]. Under UV irradiation, photosynthesis in isolated *Prochloron* cells is severely inhibited, but this process is normal in the *Prochloron* cells when it is associated with the host, which indicates that the MAAs are located in the tunic of ascidians but not in *Prochloron* cells [115]. 

Moreover, the different colors observed in the ascidians are also attributed to the presence of cyanobacteria and the variety of their secondary metabolites. The ascidian *Didemnum molle* bearing the *Prochloron* sp. exhibits great color varieties, from bright white in shallow sites to dull white in deep sites. This variation in colony color is also related to the presence of MAAs, which is essential for the photoadaptation of host ascidian [116].

*Prochloron* sp. also synthesizes the most abundant lipids, including terminal olefin lipids, nonadec-1-ene, and derivatives, by polyketide synthase (PKS) gene *gaz.* These lipids may impact the structure and the chemical components of membranes of the host animals [117,118].

The study of *Prochloron* sp. genetic modules for cyanobactin biosynthesis sets an example in understanding the pathway that evolved in the production of natural compounds. *Prochloron* sp. possesses genes for the synthesis of toxic cyanobactins, a group of ribosomally synthesized and postranslationally modified peptides (RiPPs) [119,120]. Numerous cyclic peptides, especially the patellamide class, that are secondary metabolites with pharmaceutical interest have been isolated from *Prochloron* sp. [70].

Though analysis of the patellamide biosynthetic genes, the *pat* gene cluster was identified. Seven coding sequences, *patA–patG,* are responsible for patellamide biosynthesis, all of which are transcribed in the same direction (Figure 16). Among them, *patA*, *patD, patE*, and *patG* are essential for patellamide biosynthesis [121,122]. The *patE* encodes a peptide of 71 amino acids (aa). The first 37 aa are proposed to serve as a leader sequence. As for the remaining 34 aa, 16 of them directly encode part of patellamide A and C, whereas the additional 18 aa make up motifs that may direct cyclization [123]. Within *patE*, the start and the stop recognition sequences flanking the coding regions are responsible for recruiting modifying enzymes [124,125]. *patA,* with a proline-rich coding region, encodes enzymes involved in cleavage of the patE precursor. *patD* is responsible for two domains: the N-terminal domain (patD1), which shows an adenylating enzyme activity to activate cleaved patellamide precursors as adenylates, and the C-terminus domain (patD2), which serves as a hydrolase and is involved in the cyclization of patE [121]. As for patG, which has a multidomain, its N-terminal is homologous to NAD(P)H oxidoreductases (patG1). The C-terminal patG2 contains subtilisin-like protease, indicating that patG is involved in the oxidation and the maturation of patE [123]. However, the roles of patB, patC, and patF in biosynthesis of patellamides are still not clear.

Beside this, two new cyanobactin pathways have also been discovered in *Prochloron* sp. A *tru*-like cluster, *trf*, encoding patellins 3 and 5, while a *pat*-like cluster, *bis*, encoding bistratamides A and E [126], is highly similar to the proteins that are encoded in the *pat* pathway [123,124].

### 3.3. Actinomycetes

As well-known producers of secondary metabolites such as tetracyclines, aminoglycosides, and macrolides, actinomycetes have been reported to be symbionts for different ascidians [127,128]. The genus *Streptomyces* is known as the most diverse actinomycete that widely exists in the ascidians *E. turbinata* and *Molgula manhattensis*, respectively (Figure 17) [129,130]. By analyzing the overall bacterial communities of the ascidian *Eudistoma toealensis,* the genera *Salinispora* and *Verrucosispora* were found to be the most notable colonies in the host [131]. These two actinomycetes are known for their production of indolocarbazole, which also has been isolated from the ascidian *L. patella* and enables it to be cultured in vitro [103]. A new taxonomic variation in the genus of *Gordonia*, named *Gordoniadidemni* sp. nov., has been isolated from the ascidian *Didemnum* sp. and is classified as a novel species of the genus *Gordonia* [132]. This strain exhibits the potential in bioremediation and the biodegradation of pollutants [133]. The genus *Aeromicrobium* sp., discovered from the ascidian *Halocynthia roretzi,* is the first ascidian originating symbiont that could biosynthesize taurocholic acid, a bile acid that is usually produced by mammalian liver cells [134,135]. The strain *Actinomadura* sp., associated with the ascidian *E. turbinate,* has been described to harbor an ionophore antibiotic, ecteinamycin, which has demonstrated potent activity against the strain *Clostridium difficile* in detoxification and cell death via potassium transport dysregulation [136]. Novel brominated analogues have also been isolated from such strains and have displayed potent nuclear factor E2-related factor antioxidant response element (Nrf2-ARE) activation, which is an important therapeutic approach in the treatment of neurodegenerative diseases [137]. By investigating the actinomycetes associated with three different Australian ascidians, *Symplegma rubra*, *Aplidium solidum*, and *Polyclinum vasculosum*, the genera *Streptomyces* and *Micromonospora* were highly diverse microbes, which presented rich sources for natural product discovery [138]. The fermentation extract obtained from strain *Streptomyces* sp. (USC-16018) derived from the ascidian *Symplegma rubra* yielded two polyketides, herbimycin G and elaiophylin, which showed antiplasmodial activities against chloroquine sensitive (3D7) and chloroquine resistant (Dd2) *Plasmodium falciparum* strains [139].

Numerous novel bioactive compounds have been isolated from marine-derived actinomycetes [140]. In general, most actinomycetes do not require seawater or salt supplementation for growth, but they are commonly associated with marine animals [141], where they contribute significantly to the turnover of complex biopolymers and antibiotics [142,143]. The toxic compounds produced by actinomycetes protect the host from predation and infection in shallow-water habitats [24].

### 3.4. Fungi

Compared with other microbes, fungi identified from ascidians represent only a small percentage of the total microbe community (Figure 17). The majority of them belong to the genera *Penicillium* and *Aspergillus* [144,145]. *Penicillium verruculosum* TPU1311 and *Penicillium albobiverticillium* TPU1432 isolated from Indonesian ascidians produce pharmacologically active compounds with inhibitory activities against protein tyrosine phosphatase 1B (PTP1B) [146,147]. Penicamide A, an alkaloid from the ascidian *Styela plicata*-derived *Penicillium* sp. 4829, inhibits the production of NO in RAW264.7 cells [148]. There are also reports regarding the bioactivities of compounds isolated from the ascidian *Didemnum* sp.-derived fungus *Penicillium* sp. CYE-87, which shows antimigratory and antiproliferation activity against several human cancer cell lines [149,150]. The fungus *Aspergillus* sp. KMM 4676 is associated with an unidentified ascidian from Shikotan Island [151]. A new alkaloid, asperindole A, produced by this fungus, exhibits cytotoxic activity against hormone therapy-resistant cancer cells and induces apoptosis [151]. The specificity of fungal communities has been discovered in the ascidian *Eudistoma vannamei*, in which the fungus *Aspergillus* sp. and another ten kinds of fungal strains resided. Three of the mycelium extractions of the fungal strains showed pronounced cytotoxic effects [152]. Transcriptome-based analyses showed the presence of the fungal strain *Talaromyces* sp. in ascidians collected from both Australia and South China [153,154]. This fungus produces talarolide A, the second cyclic peptide discovered from the genus *Talaromyces*, and has been proven to be a new cyclic heptapeptide rich in D-amino acids and a rare hydroxamate residue.

Fungi isolated from ascidians have been demonstrated to be capable of producing novel defense chemicals that are considered to play roles in the survival of ascidians [155], and they are involved in host–fungus and fungus–microbe interactions [156].

## 4. Concluding Remarks

Ascidians possess numerous intriguing features, such as producing abundant secondary metabolites and occupying the closest phylogenetic position to vertebrate. Recent works have revealed that plenty of bioactive compounds originate from ascidian symbionts. These compounds serve as one of the richest sources contributing to the defense infection/predation and the absorption of nutrients that are apparently necessary for the survival of the hosts. The hosts also provide amino acids, lipids, or recycled nitrogen for growth and residence of the symbiotic microbes within the ascidians. Their interactions benefit the survival and the co-evolution of both the symbionts and the host ascidians in environmental adaption (Figure 17). 

## Figures and Tables

**Figure 1 marinedrugs-17-00670-f001:**
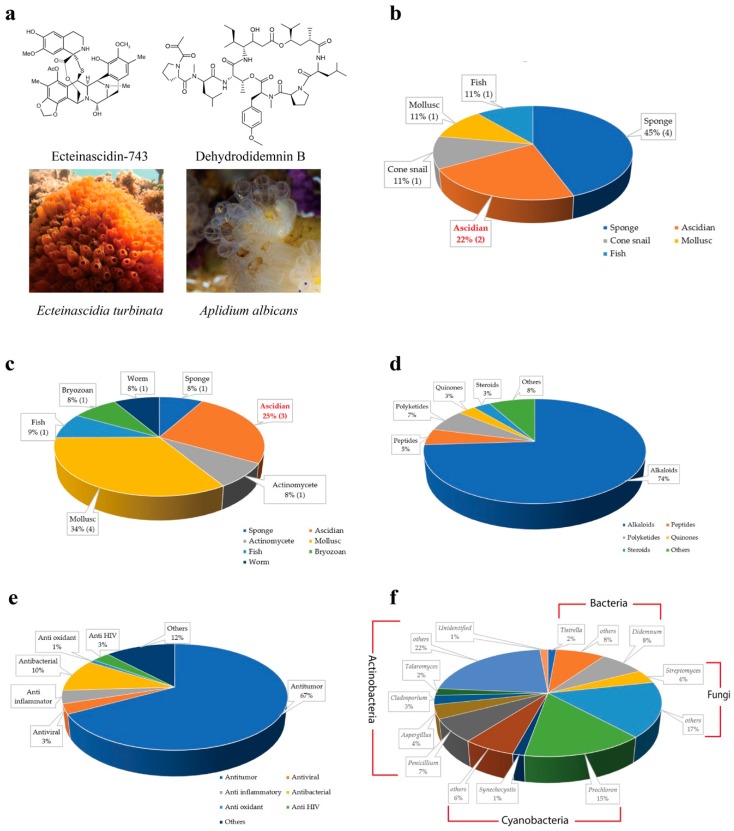
Overview of natural compounds from ascidian and the associated symbionts. (**a**) Compounds (top panels) produced by the symbiotic microbes in host ascidians (lower panels) have been approved as anti-cancer drugs or are in ongoing clinical trials by the Food and Drug Administration (FDA). (**b**) Distribution of FDA-approved drugs from different marine organisms. (**c**) Distribution of FDA-approved molecules in preclinical trials from different marine organisms. (**d**) Distribution of chemical classes of ascidian-originated compounds. (**e**) Distribution of bioactivities of natural compounds from ascidians. (**f**) Distribution of symbiont genera associated with ascidians.

**Figure 2 marinedrugs-17-00670-f002:**
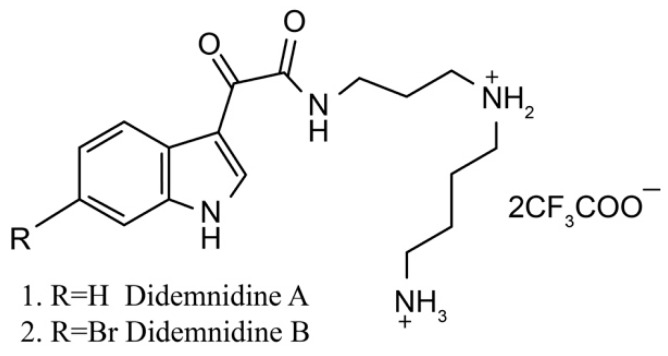
Chemical structure of didemnidines.

**Figure 3 marinedrugs-17-00670-f003:**
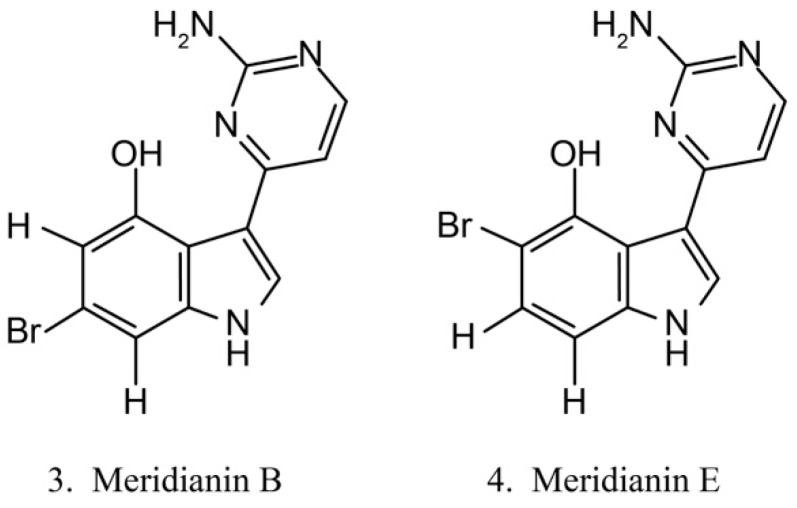
Chemical structures of meridianins.

**Figure 4 marinedrugs-17-00670-f004:**
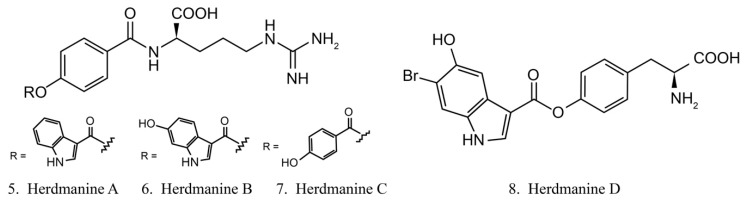
Chemical structures of herdmanines.

**Figure 5 marinedrugs-17-00670-f005:**
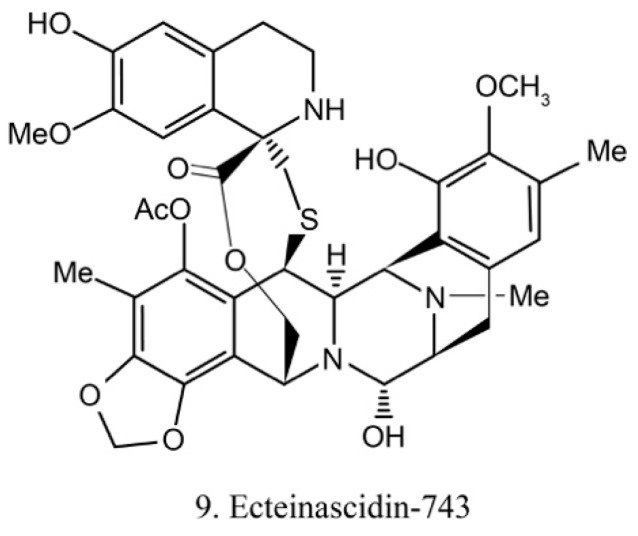
Chemical structure of ecteinascidin-743 (ET-743).

**Figure 6 marinedrugs-17-00670-f006:**
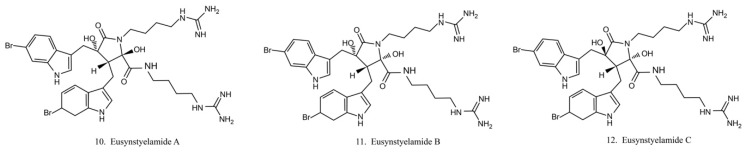
Chemical structures of eusynstyelamides.

**Figure 7 marinedrugs-17-00670-f007:**
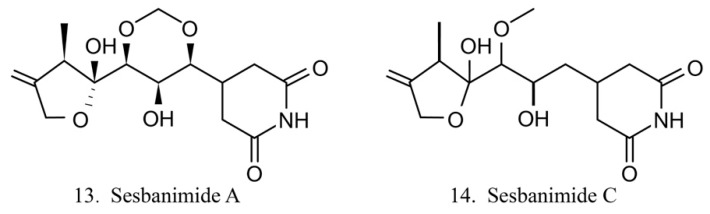
Chemical structures of sesbanimides.

**Figure 8 marinedrugs-17-00670-f008:**
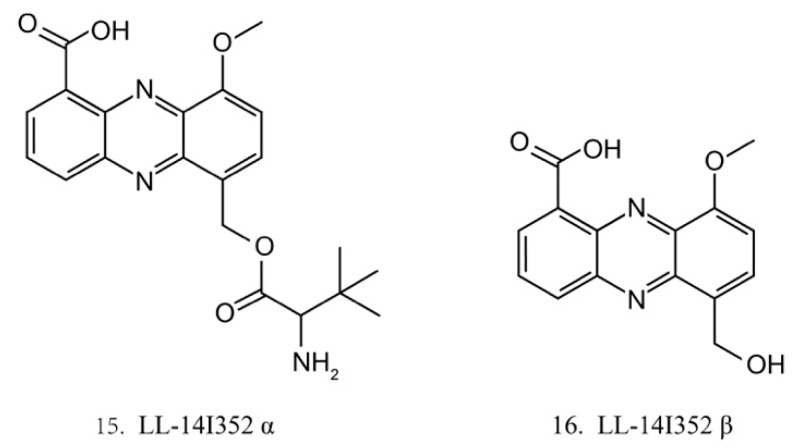
Chemical structures of LL-14I352 α and β.

**Figure 9 marinedrugs-17-00670-f009:**
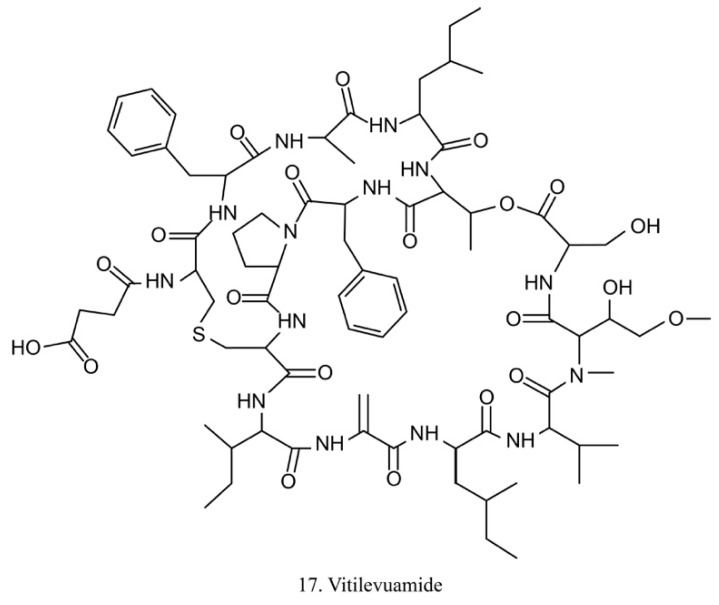
Chemical structure of vitilevuamide.

**Figure 10 marinedrugs-17-00670-f010:**
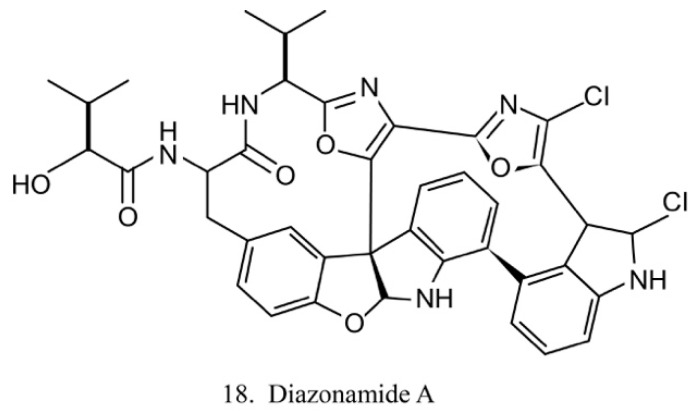
Chemical structure of diazonamide A.

**Figure 11 marinedrugs-17-00670-f011:**
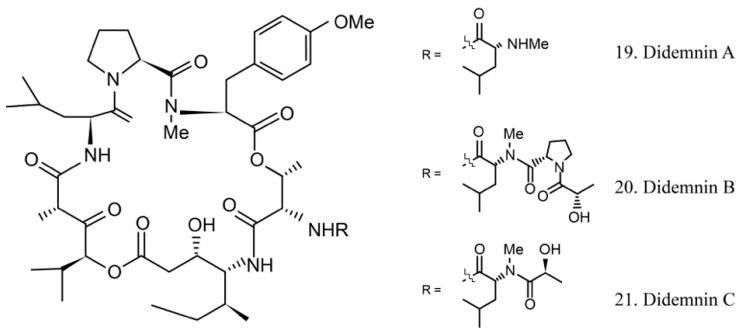
Chemical structures of didemnins.

**Figure 12 marinedrugs-17-00670-f012:**
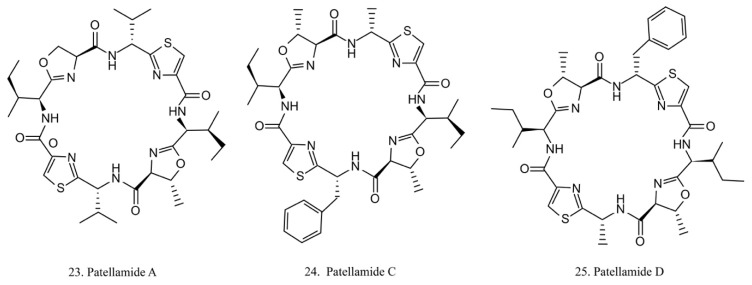
Chemical structures of patellamides.

**Figure 13 marinedrugs-17-00670-f013:**
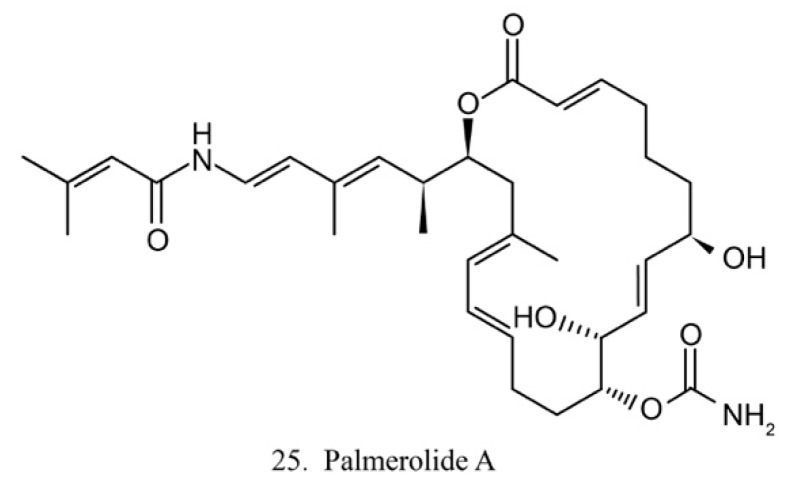
Chemical structure of palmerolide A.

**Figure 14 marinedrugs-17-00670-f014:**
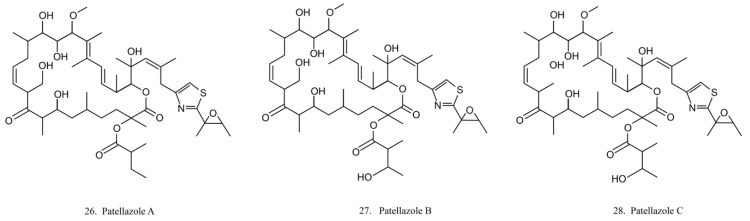
Chemical structures of patellazoles.

**Figure 15 marinedrugs-17-00670-f015:**
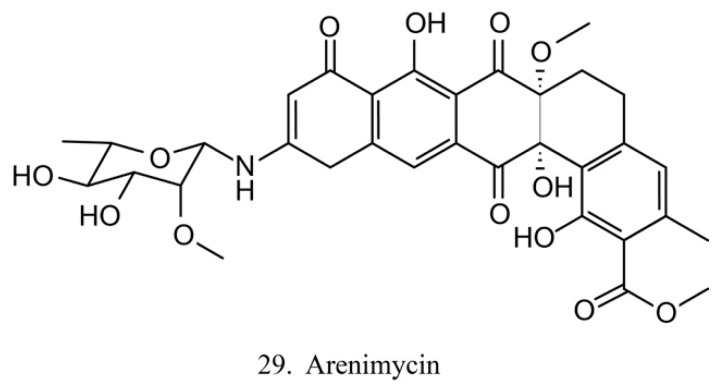
Chemical structure of arenimycin.

**Figure 16 marinedrugs-17-00670-f016:**
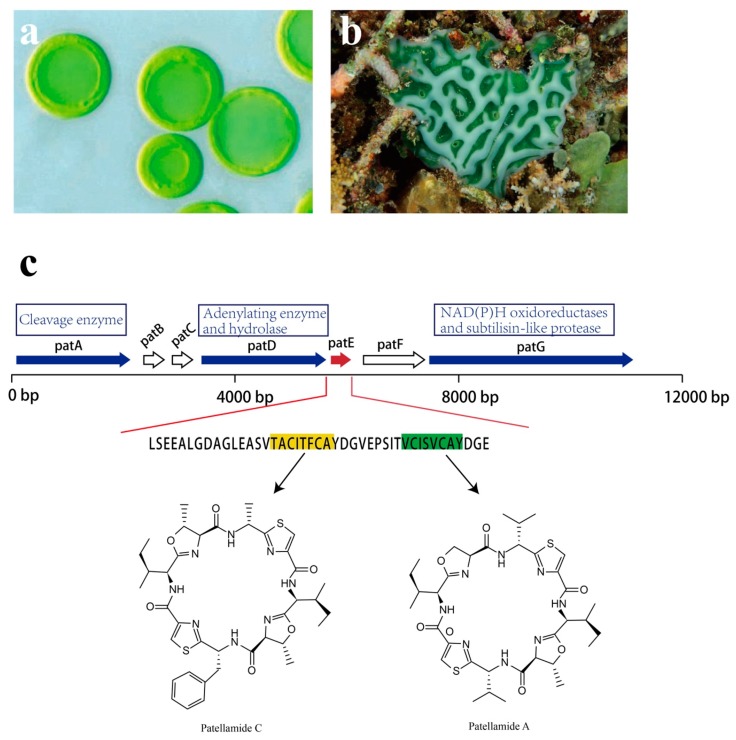
Ascidian *Lissoclinum patella*, the associated *Prochloron* sp. and *pat* pathway. (**a**) Cyanobacteria *Prochloron* sp. (**b**) ascidian *Lissoclinum patella*. (**c**) Patellamides synthesis pathway. The *pat* cluster includes seven coding sequences: *patA–patG*. *petE* (red arrow) is an essential gene for the production of patellamide C [highlighted amino acid (aa) in yellow] and patellamide A (highlighted aa in green). *PatA, patD,* and *patG* encode enzymes that are responsible for the production of patellamide (blue arrows); the functions of *patB*, *patC*, and *patF* are not clear (white arrows).

**Figure 17 marinedrugs-17-00670-f017:**
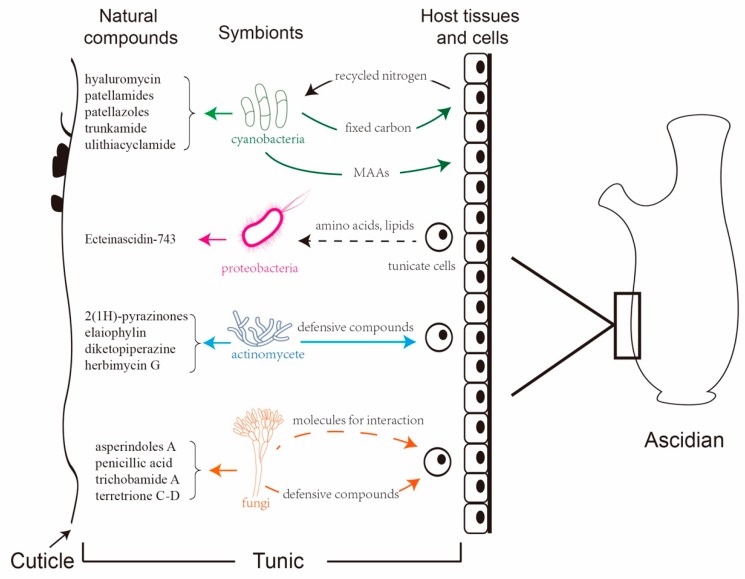
Symbionts interact with the host ascidian by producing natural compounds.

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
