# Peer review of "Origins and Bioactivities of Natural Compounds Derived from Marine Ascidians and Their Symbionts"

_marinedrugs, 2019, doi:10.3390/md17120670_

Round 1

Reviewer 1 Report

A review dealing with compounds from marine ascidians and some symbiotic microbes is proposed by Dong and Dou. The authors describe structures and activities of alcaloids, peptides, polyketides. Unfortunately, this review lacks from homogeneity (the size of the structures are different) through the all paper and the weakness of the scientific English language makes the reading difficult (many paragraphs started with very similar sentences, some other sentences are not ended, data should be better given using past tense (preterit) instead of present, etc.).

Author Response

Reviewer 1

A review dealing with compounds from marine ascidians and some symbiotic microbes is proposed by Dong and Dou. The authors describe structures and activities of alcaloids, peptides, polyketides. Unfortunately, this review lacks from homogeneity (the size of the structures are different) through the all paper and the weakness of the scientific English language makes the reading difficult (many paragraphs started with very similar sentences, some other sentences are not ended, data should be better given using past tense (preterit) instead of present, etc.).

Response:

We have re-organized and made the uniform size of the chemical structures in all the figures of the manuscript. Please see the new version of manuscript.

The English language of the manuscript has been critically checked and edited by MDPI English editing service.

Reviewer 2 Report

Review Report

Title: Origins and bioactivities of nature compounds from marine ascidians and the symbiotic microbes

The paper describes the review of the origins and biological activities of compounds isolated from marine ascidians and their symbiotic microbes.

Broad comments:

The title needs to be modified to read ‘Origins and bioactivities of natural compounds derived from marine ascidians and their symbiotic microbes’ The paper selectively discusses some of the known compounds of different structural classes, but lacks ‘punch’. This can be improved by providing some data in the introduction in the form of graphs that may be of interest to readers like: Contributions of ascidians to total number of compounds from marine in clinical use and/or pre-clinical evaluation. Out of the 1,200 compounds from ascidians, how many belong to various structural classes e.g. how may peptides, alkaloids etc. This can be discussed under various headings as found in the paper. Out of the 1,200 compounds from ascidians how many likely originated from associated microbes, and if possible break these down to bacteria/fungi followed by genus/species.

Specific comments

Specific comments are given in the ‘Marked-up’ manuscript.

Author Response

Reviewer 2

Review Report

Title: Origins and bioactivities of nature compounds from marine ascidians and the symbiotic microbes

The paper describes the review of the origins and biological activities of compounds isolated from marine ascidians and their symbiotic microbes.

Broad comments:

The title needs to be modified to read ‘Origins and bioactivities of natural compounds derived from marine ascidians and their symbiotic microbes’

Response:

The title has been changed.

The paper selectively discusses some of the known compounds of different structural classes, but lacks ‘punch’. This can be improved by providing some data in the introduction in the form of graphs that may be of interest to readers like: Contributions of ascidians to total number of compounds from marine in clinical use and/or pre-clinical evaluation. Out of the 1,200 compounds from ascidians, how many belong to various structural classes e.g. how may peptides, alkaloids etc. This can be discussed under various headings as found in the paper.

Response:

Thanks for the suggestion. We now add the data on the distribution of ascidian compounds in drugs/preclinical trails and illustrate the percentage of structural/functional classification, and prokaryotic/eukaryotic-originated classification (Figure 1b, c, d, e). The percentage and amount of each chemical classes has been added in the corresponding sections in the text.

Out of the 1,200 compounds from ascidians how many likely originated from associated microbes, and if possible break these down to bacteria/fungi followed by genus/species.

Response:

According to literature, about 100 compounds are symbiont-originated. We have added the data in the revised manuscript. The categorize the microorganisms by their genus is now present in Fig. 1f.

Specific comments

Specific comments are given in the ‘Marked-up’ manuscript.

Response:

We have corrected the manuscript based on these comments. Thanks a lot.

Round 2

Reviewer 1 Report

The referee appreciates the efforts made to have really improved the manuscript. However, the representation of the molecules still suffers from some non-homogeneity.

Author Response

We have reorganized the chemical structures in Fig. 1 and Fig. 16.